

# COVID-19 induced birth sex ratio changes in England and Wales

Gwinyai Masukume[1], Margaret Ryan[2], Rumbidzai Masukume[3], Dorota Zammit[4], Victor Grech[5], Witness Mapanga[6,7] and Yosuke Inoue[8]

[1] Independent Researcher, Munster, Ireland
[2] Trinity College Dublin, Dublin, Ireland
[3] Department of Obstetrics and Gynaecology, Faculty of Health Sciences, University of the Witwatersrand, Johannesburg, South Africa
[4] National Statistics Office, Valletta, Malta
[5] Academic Department of Paediatrics, Medical School, Mater Dei Hospital, Msida, Malta
[6] Division of Medical Oncology, Department of Medicine, School of Clinical Medicine, Faculty of Health Sciences, University of the Witwatersrand, Johannesburg, South Africa
[7] Noncommunicable Diseases Research Division, Wits Health Consortium (PTY) Ltd, Johannesburg, South Africa
[8] Department of Epidemiology and Prevention, Center for Clinical Sciences, National Center for Global Health and Medicine, Tokyo, Japan

Corresponding author
Gwinyai Masukume,
parturitions@gmail.com

## ABSTRACT

**Background**. The sex ratio at birth (male live births divided by total live births) may be a sentinel health indicator. Stressful events reduce this ratio 3–5 months later by increasing male fetal loss. This ratio can also change 9 months after major population events that are linked to an increase or decrease in the frequency of sexual intercourse at the population level, with the ratio either rising or falling respectively after the event. We postulated that the COVID-19 pandemic may have affected the ratio in England and Wales.

**Methods**. Publicly available, monthly live birth data for England and Wales was obtained from the Office for National Statistics up to December 2020. Using time series analysis, the sex ratio at birth for 2020 (global COVID-19 onset) was predicted using data from 2012–2019. Observed and predicted values were compared.

**Results**. From 2012–2020 there were 3,133,915 male and 2,974,115 female live births (ratio 0.5131). Three months after COVID-19 was declared pandemic (March 2020), there was a significant fall in the sex ratio at birth to 0.5100 in June 2020 which was below the 95% prediction interval of 0.5102–0.5179. Nine months after the pandemic declaration, (December 2020), there was a significant rise to 0.5171 (95% prediction interval 0.5085–0.5162). However, December 2020 had the lowest number of live births of any month from 2012–2020.

**Conclusions**. Given that June 2020 falls within the crucial window when population stressors are known to affect the sex ratio at birth, these findings imply that the start of the COVID-19 pandemic caused population stress with notable effects on those who were already pregnant by causing a disproportionate loss of male fetuses. The finding of a higher sex ratio at birth in December 2020, *i.e.*, 9 months after COVID-19 was declared a pandemic, could have resulted from the lockdown restrictions that initially spurred more sexual activity in a subset of the population in March 2020.

## INTRODUCTION

Male live births outnumber female live births at the population level (*Graunt, 1676*). The secondary sex ratio, commonly known as the sex ratio at birth (SRB), is calculated as male divided by total live births (*Grech, 2014*). The SRB may serve as a sentinel health indicator, revealing unfavourable conditions through a decline and better conditions through an increase (*Davis, Gottlieb & Stampnitzky, 1998*; *Grech & Masukume, 2016*).

In response to population-level events, the SRB may remain unchanged (*Grech & Scherb, 2021*; *Masukume & Grech, 2016*), transiently increase, or decrease. The increases or decreases have been noted to occur within two distinct time windows. The first window is 3–5 months after sudden and unanticipated stressful events such as terrorist attacks (*Bruckner, Catalano & Ahern, 2010*), the death of a well-known public person (*Grech, 2015*), or unexpected national election results (*Retnakaran & Ye, 2020*). This first window is linked to a disproportionate pregnancy loss of male fetuses, which is reflected in a lower SRB a few months after the stressful event. Male fetuses are more vulnerable to the effects of maternal stress (*Aiken & Ozanne, 2013*; *Catalano et al., 2021*; *Obel et al., 2007*), in accordance with the Trivers-Willard hypothesis, which holds that unfavourable environmental conditions might lower the ratio of males to females (*Trivers & Willard, 1973*).

The second window occurs 9 months after an event and can result in either a drop or a rise in the SRB. The SRB can decrease 9 months after a stressful event, such as a massive earthquake, and this is ascribed to a decrease in population-level sexual intercourse frequency and/or reduced sperm motility (*Fukuda et al., 2018*; *Fukuda et al., 1996*). The SRB may increase 9 months later if the event is linked to a population-level opportunity for more frequent sexual activity, such as the celebratory atmosphere during the first-ever home Fédération Internationale de Football Association (FIFA) 2010 World Cup tournament with a strong showing by the local team (*Masukume & Grech, 2015*). It has been established that SRB and the fertile phase of the menstrual cycle have a U-shaped association (*Guerrero, 1974*). The SRB would be skewed toward male births because of this U-shaped association if coitus was more common at the population level and more conceptions occurred further from the centre of the fertile period. The average human pregnancy lasts 9 months (*Jukic et al., 2013*).

It has been suggested that an SRB decrease could be brought on by COVID-19 stress (*Abdoli, 2020*). Indeed, we noted that the SRB fell in South Africa in June 2020, 3 months after COVID-19 was proclaimed a pandemic (*Masukume et al., 2022*), and in Japan in December 2020, 9 months after the proclamation (*Inoue & Mizoue, 2022*).

March 2020 was a key month for COVID-19 in England and Wales. This month saw an increase in local and international media coverage of the virus (*Ng, Chow & Yang, 2021*) as the World Health Organization (WHO) declared COVID-19 to be a pandemic

(*Cucinotta & Vanelli, 2020*). The first COVID-19 related death in England was also reported in March 2020 (*Mahase, 2020*). Furthermore, on 23 March 2020, the Prime Minister of the United Kingdom (UK), *i.e.,* England, Wales, Scotland and Northern Ireland made a major announcement on 'lockdown' restrictions, the closure of non-essential enterprises and the requirement to stay at home save for a few limited circumstances, in one of the most watched broadcasts. Higher levels of COVID-19-related anxiety, depression and trauma symptoms were observed in the general population in the UK from 23 to 28 March 2020 compared to previous epidemiologic periods (*Shevlin et al., 2020*). The panic buying of different household items, but especially toilet paper, which took place in several nations, including England, was another indication of the population's distress as a result of the COVID-19 pandemic onset (*Bentall et al., 2021*). This study sought to ascertain whether the SRB in England and Wales changed in response to the COVID-19 pandemic, as well as when this change took place. Our primary hypothesis was that a brief drop in the SRB occurred in England and Wales in June 2020, during the 3–5 months window after March 2020. The secondary hypothesis was that in December 2020, an SRB change took place, 9 months after March 2020.

## MATERIAL AND METHODS

### Data and statistical analysis

We used publicly available, monthly live birth data for England and Wales obtained from the Office for National Statistics, which is typically the most comprehensive data source available (*Office for National Statistics, 2022*). The 108-month period (9 years) covered by the data, from January 2012 to December 2020, was consistent with the number of months examined in earlier research on the subject (*Inoue & Mizoue, 2022*; *Masukume et al., 2022*). The most recent month for which data was available was December 2020.

The source of live birth data is a legal record that is the result of a child's parent(s) or informant formally registering the birth at a registry office. The doctor or midwife who attended the birth fills out a birth notification form. The coronavirus pandemic did not influence birth notifications the same way it did on birth registrations, which were delayed. For 2020, 0.3% of birth registrations across England and Wales were still unlinked to birth notifications (*Office for National Statistics, 2021*).

The SRB may display seasonal cycles, secular trends, or a propensity to remain elevated or depressed. Because the predicted value of an autocorrelated series is not its mean, these patterns—collectively known as autocorrelation—complicate hypothesis tests. Before examining the impact of the independent variable, temporal patterns from the dependent variable are removed to account for autocorrelation. This prevents misleading connections caused by shared autocorrelation. Therefore, seasonality or other SRB patterns are taken into account in our time series analysis (*Bruckner, Catalano & Ahern, 2010*). Using data from January 2012 to December 2019, we predicted SRBs for the 2020 months because the disease now known as COVID-19 was initially notified to WHO on 31 December 2019 (*Wu & McGoogan, 2020*). We utilized an autoregressive moving average (ARMA) model with the minimum Akaike Information Criterion (AIC) (*i.e.,* autoregressive parameter [4]

and moving average parameter [4]) to fit SRBs for January 2012 to December 2019 after checking for stationarity with the Dickey-Fuller test ($p < .001$) (Table S1). The results from this model were applied to predict SRBs for 2020. The 95% upper and lower bounds of the predicted SRBs were calculated using Stata's 'predict' command and the 'mse' option (*StataCorp, 2019*).

### Ethical considerations

The data were anonymized, thus ethical approval was not required.

## RESULTS

A total of 6,108,030 live births—3,133,915 males and 2,974,115 females—were recorded during the 9-year period between January 2012 and December 2020, with an SRB of 0.5131. From 2012 through 2020, December 2020 had the fewest live births of any month with 47,291 (Fig. 1). The SRB for June 2020 was 0.5100, the lowest of any June throughout the study period. June 2020 was also the first time June had had the lowest SRB of the year. The SRB for December 2020 was 0.5171, which was the highest for any December throughout the study period. Additionally, December 2020 marked the first time the month had the highest SRB of the year (Fig. 2). The lowest yearly SRB over the 9-year period was 0.5122 observed in 2020.

There was a significant fall in the SRB of 0.5100 in June 2020 which was below the 95% prediction interval of 0.5102−0.5179. In December 2020, there was a significant rise in the SRB of 0.5171 which was above the 95% prediction interval of 0.5085−0.5162 (Fig. 3 and Table 1).

## DISCUSSION

### Principal findings

In this study, we observed that the COVID-19 pandemic affected SRB in England and Wales. We observed that December 2020 had notably the fewest live births of any month over the 108-month study period, with 47,291 live births, 9 months after the pandemic was declared in March 2020. With an SRB of 0.5100, June 2020's SRB was significantly the lowest of any June during the study period. Indeed, the lowest SRB of the year was recorded for the first time in June 2020. This SRB decline took place 3 months after the pandemic was declared. The December 2020 SRB, which was 0.5171, was significantly the highest SRB for any December throughout the study period. Indeed, for the first time in 2020, December saw the highest SRB of the year. This significant SRB change in December 2020 suggests that the significant event affected conception dynamics in March 2020, 9 months earlier, and is unlikely to be a random finding given that a typical human pregnancy lasts 9 months.

Even though it was not our main hypothesis—which was based on month-by-month analysis—the finding that 2020 had the lowest yearly SRB at 0.5122 over the course of the nine-year period was in line with the underlying stress concept. 2020, the year the COVID-19 pandemic began globally, would be expected to be the most stressful year and indeed it had the lowest SRB.

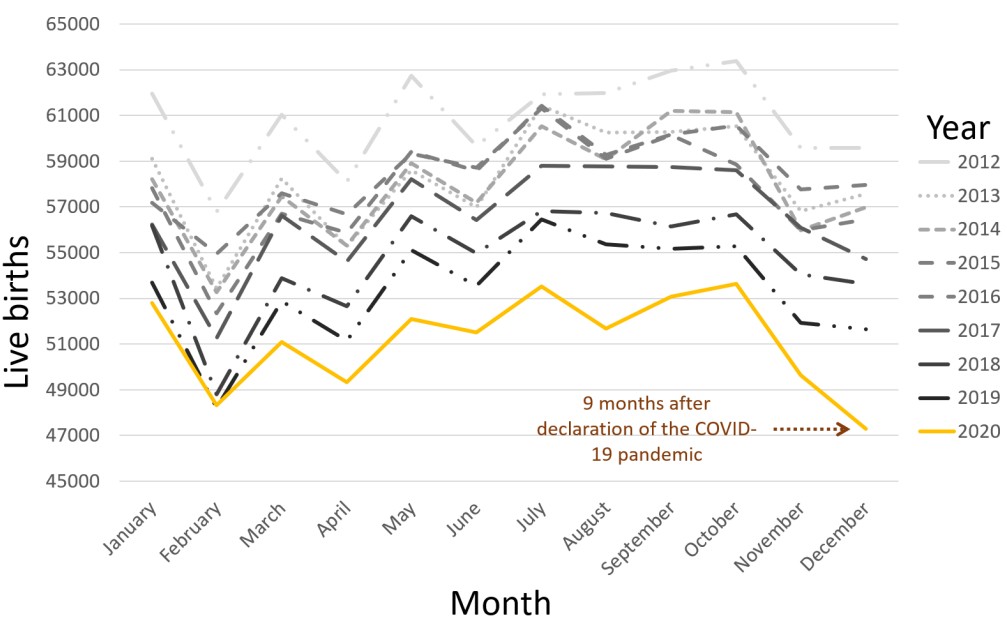

**Figure 1** Total live births over 9 years, from January 2012 to December 2020.

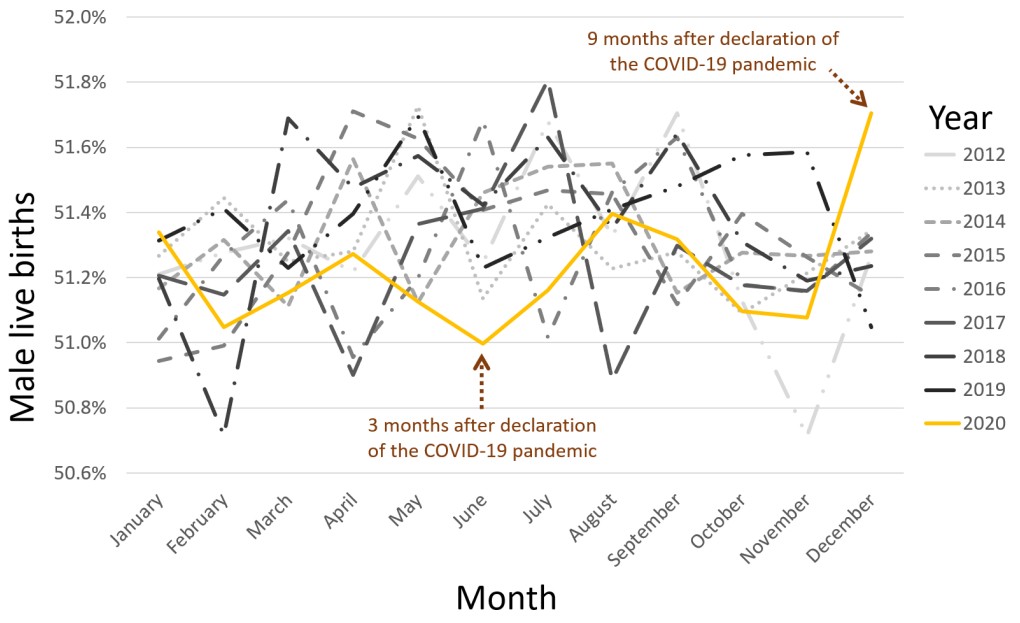

**Figure 2** Monthly proportion of male live births over 9 years, from January 2012 to December 2020.

## Comparison with other studies

Three months after the COVID-19 pandemic was declared, a decline in the SRB was noted in South Africa in June 2022 (*Masukume et al., 2022*), which is similar to the transient decline in SRB observed in this study in June 2020. This suggests that the SRB decline in
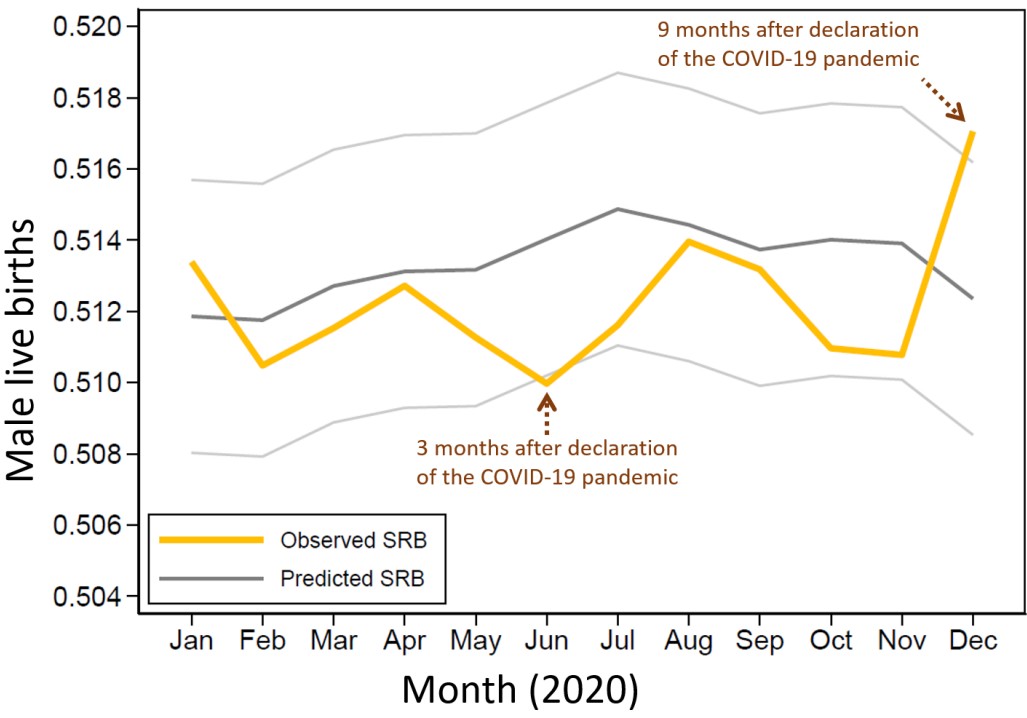

**Figure 3    Observed and predicted sex ratio at birth (SRB) in 2020.**

**Table 1    Observed and predicted sex ratio at birth for 2020.**

| Month | Males | Females | Observed SRB | Predicted SRB | Lower 95% PI | Upper 95% PI |
|---|---|---|---|---|---|---|
| January | 27,109 | 25,695 | 0.5134 | 0.5119 | 0.5080 | 0.5157 |
| February | 24,665 | 23,652 | 0.5105 | 0.5118 | 0.5079 | 0.5156 |
| March | 26,129 | 24,951 | 0.5115 | 0.5127 | 0.5089 | 0.5165 |
| April | 25,296 | 24,041 | 0.5127 | 0.5131 | 0.5093 | 0.5169 |
| May | 26,634 | 25,460 | 0.5113 | 0.5132 | 0.5093 | 0.5170 |
| June | 26,261 | 25,234 | 0.5100 | 0.5140 | 0.5102 | 0.5179 |
| July | 27,379 | 26,136 | 0.5116 | 0.5149 | 0.5110 | 0.5187 |
| August | 26,552 | 25,110 | 0.5140 | 0.5144 | 0.5106 | 0.5183 |
| September | 27,239 | 25,840 | 0.5132 | 0.5137 | 0.5099 | 0.5176 |
| October | 27,400 | 26,224 | 0.5110 | 0.5140 | 0.5102 | 0.5178 |
| November | 25,354 | 24,284 | 0.5108 | 0.5139 | 0.5101 | 0.5177 |
| December | 24,452 | 22,839 | 0.5171 | 0.5124 | 0.5085 | 0.5162 |

**Notes.**

SRB, sex ratio at birth;  PI,  prediction interval.

England and Wales in June 2020 was not coincidental but rather was being driven by a common factor that simultaneously affected different continents and hemispheres. The finding is consistent with studies that found a transient reduction in SRB 3–5 months after unanticipated occurrences that stressed populations (*Calleja, 2020*; *Catalano et al., 2006*;

*Grech, 2015*; *Retnakaran & Ye, 2020*). As previously mentioned, the unanticipated onset of the COVID-19 pandemic was identified as a substantial population stressor, and it is known that such significant stressors can induce an SRB decrease, hence it is improbable that the June 2020 SRB dip in England and Wales was a random occurrence.

Excess male fetal loss, particularly in those who were 20 to 28 weeks (4–7 months) pregnant at the time of the major stressful event, has been linked to a drop in the SRB 3 months later (*Bruckner, Catalano & Ahern, 2010*). A stillbirth is defined in England and Wales as a fetal death that occurs at ≥ 24 completed weeks of pregnancy, whereas a miscarriage happens at <24 completed weeks (*Gurol-Urganci et al., 2022*; *Jones et al., 2022*). Thus, our findings indicating a decreased SRB in England and Wales in June 2020, 3 months after the declaration of the COVID-19 pandemic, imply an increase in miscarriage and stillbirth rates during the early part of the COVID-19 pandemic, disproportionately affecting male fetuses.

Moreover, a rise in stillbirths from 2.38 per 1,000 births between 1 October 2019 and 31 January 2020 (defined as the pre-pandemic period) to 9.31 per 1,000 births between 1 February and 14 June 2020 (defined as the pandemic period) was reported from an English centre (*Khalil et al., 2020*). Although the sex of the stillbirths was not provided, in this English paper, male fetuses typically account for the majority of stillbirths (*Mondal et al., 2014*). These particular stillbirths had no connection to clinically severe acute respiratory syndrome coronavirus 2 (SARS-CoV-2) infection, which indicates an indirect stillbirth mechanism possibly involving increased maternal stress and anxiety. Internationally, miscarriages are typically not included in such statistics (*Gurol-Urganci et al., 2022*; *Jones et al., 2022*). Because a drop in the SRB might also be used to detect male fetal loss (*Bruckner, Catalano & Ahern, 2010*), our results also indicate that there may have been an unrecognised excess of male fetal loss before 24 weeks gestation (miscarriages). This is because national level statistics from England (*Gurol-Urganci et al., 2022*) and Wales (*Jones et al., 2022*) did not indicate a rise in stillbirths throughout the COVID-19 period of the current investigation, in contrast to the aforementioned single centre study (*Khalil et al., 2020*).

Despite the SRB rising, in England and Wales, in December 2020, 9 months after the COVID-19 pandemic was declared, this happened in the context of a sharp reduction in overall live births compared to Decembers prior. Contrasting this with the fact that 9 months following the 2010 FIFA World Cup in South Africa, there was an increase in SRB along with an increase in the overall number of live births (*Masukume & Grech, 2015*; *Masukume, Grech & Scherb, 2016*). This suggests that whereas a greater fraction of the population engaged in sexual activity more frequently in South Africa during the 2010 World Cup, a smaller portion of the population did so in England and Wales in March 2020. This is consistent with a British study that looked at the first 4 months of the lockdown and found that some people who were cohabiting felt like they had more partnered sexual activity since the lockdown, but people who were not cohabiting felt like they had less (*Mercer et al., 2021*). This was also in line with another British study which found an overall decrease in casual sexual activity following lockdown restrictions, while participants in serious relationships reported increases in sexual activity in comparison

to the pre-lockdown period (*Wignall et al., 2021*). A smaller portion of the population engaging in more frequent sexual intercourse would result in an increased SRB 9 months on, albeit in the context of fewer overall live births, as the likelihood of conceiving a male fetus is U-shaped with respect to the fertile period of the menstrual cycle (*Guerrero, 1974*). This might have underlain what was observed in December 2020, 9 months after the lockdown began in March 2020, in England and Wales.

Contrary to our current findings, the SRB fell in Japan, as we previously reported, in December 2020, 9 months after COVID-19 was designated a pandemic (*Inoue & Mizoue, 2022*), but this Japanese decline might have been within the same context as a significant decline in overall live births for December 2020 seen in England and Wales (*Ghaznavi et al., 2022*). This would suggest less frequent sexual encounters (*Guerrero, 1974*), and indeed declines in sexual activity associated with the start of the COVID-19 pandemic have been suggested in Japan (*Kitamura, Ae & Kosami, 2021*).

The December 2020 SRB was the highest in England and Wales compared to past Decembers, unlike in South Africa where it stayed unchanged (*Masukume et al., 2022*). The latter was in line with historical December patterns of total live births (*Statistics South Africa, 2021*), unlike in Japan (*Inoue & Mizoue, 2022*) and England and Wales. The fact that the SRB did not alter noticeably 9 months after the declaration of the COVID-19 pandemic (*i.e.,* December 2020) suggests that the population-level frequency of partnered sexual activity in South Africa in March 2020 may have not differed noticeably from earlier years.

Some of the SRB differences 9 months after March 2020, in December, between the countries in this study may be explained by different population characteristics, even though it is challenging to pinpoint the exact causes. For instance, in 2020, the estimated median population ages for Japan, South Africa and the UK respectively, were 48.0, 26.9 and 39.5 years (*Ritchie & Roser, 2019*). Although many interrelated factors affect how frequently people engage in sexual activity and thereby indirectly influencing SRB, it is known that sexual activity reduces with age (*Beutel, Stöbel-Richter & Brähler, 2008*). This may explain why South Africa, with the youngest population, had an essentially unchanged SRB while Japan with the oldest population had a low SRB in December 2020.

Overall, our research demonstrates that the SRB is a very valuable sentinel health indicator since, unlike other indicators, it is usually available for entire populations (*Davis, Gottlieb & Stampnitzky, 1998*). Population dynamics can be better understood by triangulating the SRB with additional data.

## Strengths and limitations

The study's results cannot be extrapolated to the individual level due to the ecological fallacy (*Björk et al., 2021*). However, the ecologic study method is arguably the most appropriate for examining how a population stressor affects a population outcome (*Pearce, 2011*).

March 2020 saw the temporary suspension of birth registration in England and Wales. As soon as it was safe to do so, registrations started up again in June 2020. Under normal circumstances, more than 95% of births are typically documented within the required 42-day window. In 2020, only 58% of births were reported within 42 days, a notable reduction.

Due to this known delay, the cut-off date for incorporating 2020 live birth registrations was five and a half months later than usual, in August 2021. Similar distributions for the baby's sex were found in both the linked registrations data and the unlinked notifications when birth registrations and birth notifications were compared. The aforementioned birth registration delays had little bearing on the current analysis because 0.3% of notifications across England and Wales were still unlinked (*Office for National Statistics, 2021*). Although the national lockdown was enforced concurrently, the extent of other COVID-19 prevention measures may have varied by region. Given that the publicly accessible data we used only covered the national level, it was not possible to investigate the variations in effects by region.

The analysis presented here pertains to England and Wales' initial COVID-19 wave (*Knock et al., 2021*). England and Wales had seen more waves at the time of writing. In the waves that followed, the SRB might have been perturbed. When the live birth data for these times are made accessible, this is a topic for further study. In the years between 2012 and 2020, we also noticed other major SRB dips and peaks, but we had not *a priori* hypothesised how they might have arisen; thus, we did not address them further in this paper.

## CONCLUSIONS

These results suggest that the start of the COVID-19 pandemic caused population stress with notable effects on those who were already pregnant by causing a disproportionate loss of male fetuses given that June 2020 falls within the critical window when population stressors are known to affect the sex ratio at birth. The finding of a higher sex ratio at birth in December 2020, 9 months after COVID-19 was declared a pandemic, could have resulted from the lockdown regulations that encouraged more partnered sexual activity in a subset of the population. Future pandemic preparedness (*Marston, Paules & Fauci, 2017*) and social policy (*Cook & Ulriksen, 2021*) can be influenced by our findings. Examples of how this knowledge could be applied to prevent a potential SRB reduction include targeted increases in resources for maternal health services, including improved fetal surveillance, particularly during and in the lead up to 3–5 months after the start of a future pandemic. Future studies should examine whether the SRB changed in other nations after the onset of COVID-19.

## ACKNOWLEDGEMENTS

We acknowledge the Office for National Statistics for the recorded live birth data.

### Funding

The University of Malta funded the article processing charge. The funders had no role in study design, data collection and analysis, decision to publish, or preparation of the manuscript.

## Competing Interests

The authors declare there are no competing interests. Witness Mapanga is employed by Wits Health Consortium (PTY) Ltd.

## Author Contributions

- Gwinyai Masukume conceived and designed the experiments, performed the experiments, analyzed the data, prepared figures and/or tables, authored or reviewed drafts of the article, and approved the final draft.
- Margaret Ryan conceived and designed the experiments, performed the experiments, authored or reviewed drafts of the article, and approved the final draft.
- Rumbidzai Masukume conceived and designed the experiments, performed the experiments, authored or reviewed drafts of the article, and approved the final draft.
- Dorota Zammit conceived and designed the experiments, performed the experiments, authored or reviewed drafts of the article, and approved the final draft.
- Victor Grech conceived and designed the experiments, performed the experiments, analyzed the data, authored or reviewed drafts of the article, and approved the final draft.
- Witness Mapanga conceived and designed the experiments, performed the experiments, authored or reviewed drafts of the article, and approved the final draft.
- Yosuke Inoue conceived and designed the experiments, performed the experiments, analyzed the data, prepared figures and/or tables, authored or reviewed drafts of the article, and approved the final draft.

## Human Ethics

The following information was supplied relating to ethical approvals (i.e., approving body and any reference numbers):

The data were anonymized, thus ethical approval was not required.

## Data Availability

The data is available at the Office for National Statistics: https://www.ons.gov.uk/peoplepopulationandcommunity/birthsdeathsandmarriages/livebirths/adhocs/15002livebirthsbymonthofoccurrenceandsexofbabyenglandandwales2012to2020.

## Supplemental Information

Supplemental information for this article can be found online at http://dx.doi.org/10.7717/peerj.14618#supplemental-information.

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
