# Peer review of "COVID-19 induced birth sex ratio changes in England and Wales"

_PeerJ, doi:10.7717/peerj.14618_

## Round 0.1 · original submission · Major Revisions

Based on detailed reviewers comments, authors are invited to address the concerns raised by reviewers. Authors should focus on validity of their findings and methodological concerns such as study heterogeneity, use of time series analysis.

Reviewer 1 ·

Basic reporting

no comment

Experimental design

no comment

Validity of the findings

I am not sure whether the low SRB 3 months after covid-19 pandemic and the high SRB 9 months since the pandemic are really due to the stressful environment and increased sex intercourse on the population level. When looking at Figure 2, without highlighting the 2020 SRB, I cannot really see much difference between 2020 and other years' SRB time trends. Although the 3 and 9 month SRB are outside the 95% prediction interval (PI), but according to Figure 3, they barely pass the 95% PI. If the author do this exercise again to generate PI for the year 2019 using data from 2012-2018, there might be SRB in one or more months fall outside the 95% PI. Furthermore, since it is called "95% prediction interval", it is expected that around 5% of the left-out data (in this case the monthly observed SRB in 2020) will fall outside the 95% PI. 5% of 12 data points is 0.6 data point. Here 2 data points are outside. So maybe at least 1 data point is meant to fall outside regardless the occurrence of the covid-19 pandemic. Last but not the least, as shown in Figure 2, the total number of live births in 2020 are in general lower than previous years. Hence, given a smaller population size in 2020, the stochastic uncertainty is very possible to be larger purely due to a small population size.

Additional comments

If the authors could discuss more on whether the identified "outliers" are contributed more by the covid-19 pandemic or rather mainly due to natural fluctuation, it will be more beneficial to the readers.

Reviewer 2 ·

Basic reporting

Sex ratio at birth (SRB) is a measure that may capture the impact of the event after a certain period it occurred. Like Japan and South Africa, it is reasonable that many researchers focus on the SRB in specific countries. The authors tried to argue that COVID-19 strongly impacted people who were supposed to have babies in England and Wales in 2020. The manuscript was well written, but there are some comments that I would like the authors to consider.

Experimental design

no comment

Validity of the findings

Please provide the figure of the fitting result (learning period from 2012 to 2019).

Is it possible to explore the geographical heterogeneity of the SRB in England and Wales? Although the national lockdown may have been implemented simultaneously, other measures against the transmission of COVID-19 and their degree could have differed from each region. So that would be nice if the authors gave some insights into the geographical heterogeneity of SRB.

I propose that the authors discuss the pros and cons of using the time-series analysis for SRB. Additionally, please validate the method of application of time series if the authors know of any previously published scientific papers comparing the estimated predicted from the previous trend of and observed SRB.

Additional comments

no comment

---

## Round 0.2 · Minor Revisions

Dear authors, there are afew minor comments that need your attention. Kindly rephrase your conclusion and address reviewer comments.

Reviewer 1 ·

Basic reporting

Fine.

Experimental design

Fine in general but see my additional comments below.

Validity of the findings

See my additional comments below.

Additional comments

General comments

I appreciate the effort that the authors put into replying to my previous comments. However, I am still not fully convinced of the conclusions they draw based on the monthly SRB trends. That said, I do not deny the possibility that the covid pandemic did influence SRB. The paper is still valuable by presenting the observed monthly SRB with authors' speculations. It is informative to present this possibility (which is not 100% certain) to the readers.


Figure 2: the authors put more explanations. But, the observed SRB 3 and 9 months after the declaration of covid pandemic don’t look that outlying and both are within the uncertainty of natural fluctuations. E.g. the lowest SRB is in Feb 2015 (or 2016) and the highest SRB is in July 2014 (or maybe 2015, I can’t tell clearly from the legend). Hence, if those minimum and maximum SRB values do not have any special meaning and are considered as natural fluctuations, why should we think the SRB 3 and 9 months after the declaration of the covid pandemic is any special? Also, if the monthly SRB is available for more years back, the author can add those time trends in the same plot and see the SRB fluctuations for a longer period.

Reviewer 2 ·

Basic reporting

no comments

Experimental design

no comments

Validity of the findings

no comments

Additional comments

no comments

---

## Round 0.3 · accepted · Accept

The authors have addressed all the comments sufficiently.